# Resource utilization and outcomes in emergency general surgery during the COVID19 pandemic: An observational cost analysis

**Amelia J. Hessheimer**[1]*, **Marta Trapero-Bertran**[2], **Alex Borin**[3], **Eugenia Butori**[3,4], **Anna Curell**[4], **Arlena Sofía Espinoza**[3,4], **Joaquín Jensen**[3], **Víctor Turrado**[4], **Xavier Morales**[4], **Antonio María de Lacy**[4], **Constantino Fondevila**[1]

1 General & Digestive Surgery, Institut de Malaties Digestives i Metabòliques (ICMDM), Hospital Clínic Barcelona, IDIBAPS, CIBERehd, University of Barcelona, Barcelona, Spain, 2 Basic Sciences Department, University Institute for Patient Care, Universitat Internacional de Catalunya Barcelona, Barcelona, Spain, 3 General & Digestive Surgery, ICMDM, Hospital Clínic Barcelona, Barcelona, Spain, 4 Gastrointestinal Surgery, ICMDM, Hospital Clínic Barcelona, Barcelona, Spain

* HESSHEIMER@gmail.com

**Data Availability Statement:** The minimal anonymized data set for this study may be

## Abstract

### Background

Over the course of the COVID19 pandemic, global healthcare delivery has declined. Surgery is one of the most resource-intensive area of medicine; loss of surgical care has had untold health and economic consequences. Herein, we evaluate resource utilization, outcomes, and healthcare costs associated with unplanned surgery admissions during the height of the pandemic in 2020 versus the same period in 2019.

### Methods

Retrospective analysis on patients ≥18 years admitted from the emergency department to General & Digestive and Gastrointestinal Surgery Services between February and May 2019 and 2020 at our center; clinical outcomes and unadjusted and adjusted per-person healthcare costs were analyzed.

### Results

Consults and admissions to surgery declined between February and May 2020 by 37% and 19%, respectively, relative to the same period in 2019, with even greater relative decline during late March and early April. Time between onset of symptoms to diagnosis increased from 2±3 days 2019 to 5±22 days 2020 ($P = 0.01$). Overall hospital stay was two days less in 2020 ($P = 0.19$). Complications (Comprehensive Complication Index 10.3±23.7 2019 vs. 13.9 ±25.5 2020, $P = 0.10$) and mortality rates (3% vs. 4%, respectively, $P = 0.58$) did not vary. Mean unadjusted per-person costs for patients in the 2019 and 2020 cohorts were 5,886.72 €±12,576.33€ and 5,287.62±7,220.16€, respectively ($P = 0.43$). Following multivariate analysis, costs remained similar (4,656.89€±390.53€ 2019 vs. 4,938.54±406.55€ 2020, $P = 0.28$).

accessed at the Harvard Dataverse(DOI: 10.7910/DVN/UAAX8W; https://dataverse.harvard.edu/dataset.xhtml?persistentId=doi:10.7910/DVN/UAAX8W).

**Funding:** The authors received no specific funding for this work.

**Competing interests:** Amelia J. Hessheimer and Constantino Fondevila have received consultancy fees from Guanguong Shunde Innovative Design Institute, Guangdong, China, and research funding from Instituto de Salud Carlos III. This does not alter their adherence to PLOS ONE policies on sharing data and materials. The remainder of authors have no disclosures.

**Abbreviations:** AIC, Akaike information criterion; ASA, American Society of Anesthesiologists; CCI, Comprehensive Complication Index; ED, emergency department; GLM, generalized linear models; HCB, Hospital Clínic Barcelona; ICU, intensive care unit; OR, operating room.

## Conclusions

Healthcare delivery and spending for unplanned general surgery admissions declined considerably due to COVID19. These results provide a small yet relevant illustration of clinical and economic ramifications of this healthcare crisis.

## Introduction

Apart from its direct impact on infected patients, the SARS-CoV-2/COVID19 pandemic has had indirect collateral effects on patients with other pathologies not accessing standard of care for their diseases [1, 2]. Surgery is one of the most resource-intensive areas of clinical medicine [3], and patients requiring surgical interventions have been particularly vulnerable in this regard. Factors affecting surgical patients have included lack of operating room personnel and space. Inability and/or fear of patients with incidental surgical pathology to present in timely fashion has resulted in a well-documented declines in surgical consults during the pandemic in many parts of the world [4–9]. Moreover, it has been suggested that these patients may have presented later, with more advanced disease [9, 10]. These issues have potential to impact patients' lives and quality of life and put further strain on public money and healthcare resources.

Cost studies in healthcare are performed with the objective of transforming impact of a disease in economic terms. They provide empirical evidence to facilitate assessment by public decision-makers to make rational decisions regarding efficient allocation of resources.

It is unclear how the COVID19 pandemic may have changed healthcare spending on patients with incidental surgical pathology derived from the emergency department. We hypothesized that analysis of costs associated with unplanned general surgery admissions during 2020 would demonstrate differences versus those of the previous year, 2019, where no pandemic occurred. We performed an exploratory cost analysis to determine differences in resource utilization, clinical outcomes, and direct healthcare costs. We aimed to identify what patterns may have existed and if they varied according to presenting diagnosis, in order to identify areas for improving expenditures without compromising outcomes in new waves of the same disease or new pandemics of a similar nature.

## Patients and methods

### Study design

This is a retrospective study on adult patients ($\geq$18 years) admitted from the Emergency Department (ED) to the General & Digestive and Gastrointestinal Surgery Services at the Hospital Clínic Barcelona (HCB), a tertiary care center serving an encatchment population of 540,000. Patients admitted between February and May 2020 and the same period in 2019 were included.

Data was collected from the HCB Electronic Medical Record (SAP, S.E., Walldorf, Baden-Württemberg, Germany) and the Shared Medical Record of the Catalan Department of Health. Data collection did not violate any of the usual data provider processes. This study was conducted in compliance with the Declaration of Helsinki (current version October 2013, Fortaleza, Brazil) and in accordance with existing protocol and legal requirements ("Boletín Oficial del Estado Orden SAS/3470/2009", December 16, 2009). Study approval was obtained

from the HCB Committee on Ethics in Medical Research prior to data collection (study number HCB/2020/0542), which waived need for written consent.

## Definition of variables and outcomes

The main objectives of the study were to assess clinical outcomes and costs associated with general surgery ED admissions during the peak of the pandemic in 2020 relative to the same period in 2019. The study was conducted from the perspective of the hospital provider; no primary care or other ambulatory healthcare costs were imputed.

**Clinical variables and outcomes.**   The following baseline sociodemographic and clinical variables were recorded: age, sex, country of birth, civil status, level of education, profession, employment situation, American Society of Anesthesiologists (ASA) classification [11], and history of previous surgery. Variables related to admission included principal diagnosis; date of diagnosis; duration of symptoms prior to diagnosis; and non-surgical treatment, when applicable, and success or failure of such treatment. Surgical variables, when applicable, included date of surgery, intraoperative complications, and operative time. Hospitalization variables included length of stay and, when applicable, intensive care unit (ICU) stay; surgical reintervention; mortality during index admission or up to 30 days thereafter; and the Comprehensive Complication Index [12], which computes all complications a patient experiences in an index ranging from 0 to 100. Diagnosis of COVID19 was recorded for patients in the 2020 cohort.

Admissions were grouped into the following diagnostic categories: appendicitis (acute forms, including appendicular infiltrate), biliary tract pathology (acute cholecystitis, cholangitis, choledocholithiasis, and other forms of obstructive jaundice), gastrointestinal obstruction (partial and complete obstruction due to adhesive disease, herniation, neoplasm, volvulus, etc.), non-obstructive gastrointestinal pathology (including acute diverticulitis, colitis, and gastrointestinal tract perforation of any cause), skin and soft tissue infection (including perianal and other superficial abscesses and necrotizing fasciitis), abdominal trauma (both blunt and penetrating), and readmission (complications arising as a direct consequence of a previous surgical intervention or within 30 days of discharge). Admissions falling outside these categories were categorized as other.

**Costs.**   All costs are expressed in 2020 Euros (€). Costs associated with index admission and readmissions with 30 days of discharge were obtained from the Hospital Clínic. Costs of hospitalization included pharmacy, transfusions, artificial implants (stent, mesh, etc.), miscellaneous consumable materials, physician medical care, general floor stay, studies and procedures (radiological, endoscopic, etc.), laboratory tests, consultations from other specialists (physiotherapy, nutritional services), and sanitation/janitorial services. Operative costs included consumables (anesthetic drugs, transfusions, surgical instrumentation, and devices) and operating room (OR) and staff costs. Death did not result in cost-censoring and was considered to represent complete data in that no further cost was accrued; costs of patients who died before discharge were included in the analysis.

## Data presentation and analysis

Unadjusted and adjusted direct healthcare costs are presented for the four-month periods in 2019 and 2020. Continuous variables are presented as mean ± standard deviation and categorical variables as frequencies, unless otherwise specified. Univariate analysis of costs and differences in continuous and categorical variables were assessed using the parametric test of cost on untransformed (raw) scale, univariate two-sample *t* test, and Chi-squared test, respectively. Multivariate analysis of costs was performed to adjust for potential cost variation due to other causes (confounding variables). Generalized linear models (GLM) were used for multivariate

analysis, as they allow for heteroscedasticity through a variance structure, relating variance to mean [13]. The modified Parks test was used to identify the Gamma family for the GLM [14]. In certain cost categories where a large number of cases incurred no costs, a two-part model was used. The two-part model consisted in a probit model followed by GLM, with Gamma family and log link. To select the model with the best fit, we evaluated Pseudo R2 for the first part and the Akaike information criterion (AIC) for the second part of the model. Statistical significance was defined as $P{<}0.05$. Statistical analyses were performed using STATA® 16.0 (StataCorp, LLC, College Station, Texas, USA).

## Results

In the four-month periods in 2019 and 2020, there were a total of 33,654 and 23,627 consults to the HCB Emergency Department, among which 4,326 (13%) and 2,722 (12%), respectively, were triaged to general surgery (59% decrease between years). From these populations, 772 patients were ultimately admitted and included in the analysis: 426 from 02-05/2019 and 346 from 02-05/2020 (23% decreased between years).

General surgery consults and admissions from the ED were stable throughout the four-month period in 2019, with higher activity in March relative to the other three months. While consults and admissions were similar in February 2019 and 2020, both declined considerably in March 2020 relative to both March 2019 (consults -55%, admissions -48%) and February 2020 (consults -48%, admissions -41%). This decline continued through April and returned to near normal by May 2020 (Fig 1).

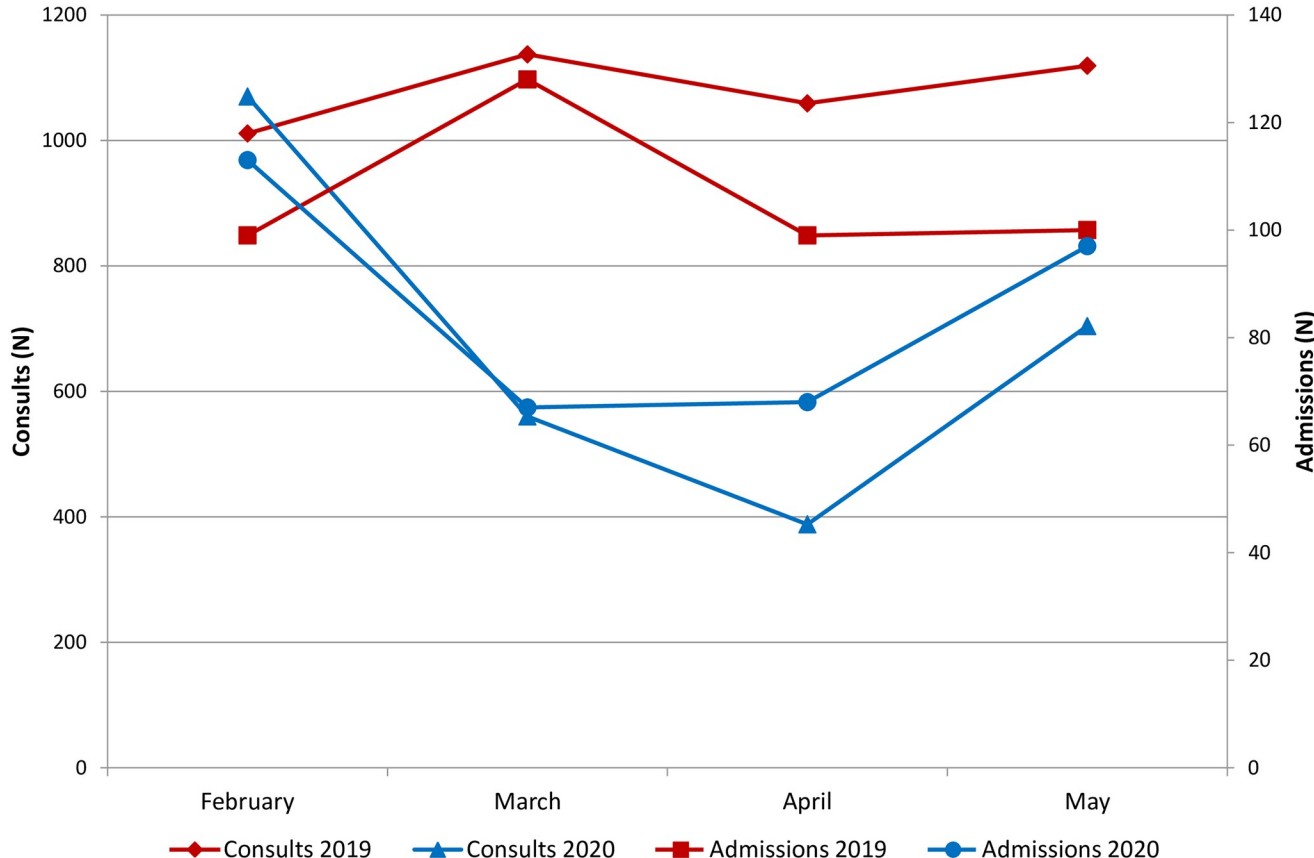

**Fig 1. Emergency surgery consults and admissions.** Monthly evolution of unplanned general surgery consults and admissions during two periods: 02-05/2019 (pre-pandemic) and 02-05/2020 (COVID19 pandemic).

## Patient demographics

Baseline patient sociodemographic and clinical characteristics are listed in Table 1. The average age of all patients was 60±20 years, 43% were women, 78% were from Spain, and 39% were married; 46% were pensioners who had previously worked and another 37% were actively employed. Age, sex, country of birth, employment situation, profession, ASA classification, and history of prior surgery did not differ between the two cohorts; civil status was unavailable for a greater proportion of case in the 2020 cohort relative to 2019.

## Clinical variables and outcomes

Table 2 provides data from the two cohorts related to admissions, interventions, and overall hospital stays. Among patients in the 2020 cohort, ten (3%) were concomitantly diagnosed with COVID19.

There were differences in the representation of the different diagnostic categories between the two periods, with the greatest number of admissions in 2019 related to acute appendicitis (N = 106, 25%) and the greatest number in 2020 related to biliary tract pathology (N = 103, 30%). Time between onset of symptoms to diagnosis was significantly increased from an average of two days in 2019 to five days in 2020 ($P = 0.01$). The percent of cases initially managed non-operatively and rate of failure of non-operative management did not vary. Intraoperative complications and operative times were similar. Overall hospital stay was, on average, two days less in 2020, though this difference was not significant. Complications (CCI 10.3±23.7 2019 group vs. 13.9±25.5 2020 group, $P = 0.10$) and mortality rates (3% vs. 4%, respectively, $P = 0.58$) did not vary.

## Cost outcomes

Costs for hospitalization, surgery, and readmissions were available for 409 patients from 2019 and 334 from 2020.

**Overall costs outcomes.** Mean unadjusted per-person healthcare costs for patients in the 2019 and 2020 cohorts were 5,886.72€±12,576.33€ and 5,287.62±7,220.16€, respectively ($P = 0.43$). Median per-person costs were lower in both instances (2,532.77€ and 3,217.40€, respectively), indicating that cost distributions skewed left and the majority of patients had costs below the arithmetic means. Fig 2 reflects how costs were distributed per month in each period.

Following unadjusted univariate analysis, a few significant differences between the two periods were noted. Higher costs associated with miscellaneous consumables used during hospitalization and lower costs associated with OR consumables and OR usage were observed in 2020 relative to 2019 (Table 3). Following multivariate analysis and adjustment of costs to account for potential influence of confounding covariates, mean per-person healthcare costs remained similar between the two periods. Marginal but statistically significant differences were observed for costs associated with pharmacy and miscellaneous consumable materials (higher in 2020) and OR consumables (lower in 2020).

**Outcomes according to diagnosis.** In the three largest diagnostic categories—appendicitis, biliary tract pathology, and gastrointestinal obstruction—there were a total of 103, 87, and 69 patients, respectively, with costs available from 2019 and 71, 101, and 45, respectively, from 2020. While 2019 and 2020 costs were the same in the latter two categories, univariate analysis reflected differences in unadjusted costs in the appendicitis subpopulation, with higher total healthcare costs in 2020 relative to 2019 (3,324.93€±175.80€ vs. 2,663.76€±96.49€, respectively, $P<0.01$). Cost differences were based in large part on higher costs for general floor stay (829.01€±78.37€ vs. 636.15€±51.55€, $P = 0.03$), physician medical care (160.68€±18.13€ vs.

**Table 1. Sociodemographic and clinical characteristics.**

| | 2019 (N = 426) | 2020 (N = 346) | *P* |
|---|---|---|---|
| Age (y) | 59±20 | 60±20 | 0.55 |
| Age range (%) | | | 0.44 |
| 18–30 | 9 | 10 | |
| 31–40 | 12 | 10 | |
| 41–50 | 17 | 13 | |
| 51–60 | 13 | 15 | |
| 61–70 | 15 | 15 | |
| 71–80 | 14 | 19 | |
| 81–90 | 15 | 15 | |
| >90 | 5 | 3 | |
| Women (%) | 41 | 45 | 0.24 |
| Country of birth (%) | | | 0.73 |
| Spain | 80 | 76.1 | |
| Rest of Europe | 3.5 | 3.2 | |
| Middle East | 0.7 | 1 | |
| Africa | 1.8 | 2.7 | |
| Latin America | 10.6 | 13 | |
| Asia | 2.8 | 2.1 | |
| North America | 0.5 | 0 | |
| N/A | 0.3 | 1.6 | |
| Civil status (%) | | | 0.001 |
| Single/no partner | 16.4 | 10.7 | |
| Married/couple | 40.6 | 36.6 | |
| Divorced/separated | 5.9 | 6 | |
| Widow/-er | 9.4 | 5.2 | |
| N/A | 27.7 | 41.5 | |
| Level of education (%) | | | 0.10 |
| Illiterate/unable to read or write | 2 | 1 | |
| Incomplete primary education | 9.4 | 8 | |
| Complete primary education | 26.8 | 22 | |
| Secondary education | 10.4 | 15 | |
| High school diploma or equivalent | 14.1 | 11.5 | |
| Associate's degree | 9.1 | 11.5 | |
| Bachelor's degree | 24.5 | 30.5 | |
| Postgraduate education | 3.7 | 0.5 | |
| Employment situation (%) | | | 0.32 |
| Employed | 37 | 36.1 | |
| Pensioner who has worked | 45.7 | 47.3 | |
| Pensioner who has not worked | 5.7 | 2.9 | |
| Unemployed | 5.3 | 8.8 | |
| Student | 2 | 1 | |
| Unpaid domestic work | 2.3 | 1.5 | |
| Other | 0.7 | 1.9 | |
| N/A | 1.3 | 0.5 | |

(*Continued*)

**Table 1.** (Continued)

| | 2019 (N = 426) | 2020 (N = 346) | *P* |
|---|---|---|---|
| Profession (%) | | | 0.50 |
| Director, manager, or administrator | 4 | 1.6 | |
| Scientific professional, intellectual | 23.5 | 29 | |
| Technician, support professional | 18 | 18.1 | |
| Accountant, office worker | 1.8 | 3.6 | |
| Food service worker, personal care provider, protection, or sales | 20 | 20.2 | |
| Skilled worker in agriculture, livestock, forestry, or fishing | 0.4 | 0 | |
| Skilled craftsperson, manufacturing, or construction worker | 16.9 | 16.1 | |
| Installation, machinery, or assembly worker | 3.3 | 3.1 | |
| Elementary or secondary education provider | 12.1 | 8.3 | |
| ASA classification (%) | | | 0.47 |
| I | 23.7 | 25 | |
| II | 52.8 | 48.5 | |
| III | 21.1 | 25 | |
| IV | 2.4 | 1.5 | |
| Previous surgery (%) | 42.5 | 38.8 | 0.29 |

Baseline sociodemographic and clinical characteristics of patients with unplanned general surgery admissions during two periods: 02-05/2019 (pre-pandemic) and 02-05/2020 (COVID19 pandemic). Continuous variables are presented as mean ± standard deviation and categorical variables as frequencies. ASA, American Society of Anesthesiologists; N/A, information not available.

102.81€±9.89€, $P<0.01$), miscellaneous consumable materials (164.65€±33.42€ vs. 2.84€±2.84 €, $P<0.01$), and OR staff (688.51€±37.07€ vs. 562.27€±25.49€, $P<0.01$). Given relatively small sample sizes and missing data among covariates, cost adjustments could not be performed. Of note, complication rates were the same in 2019 and 2020 (CCI 11.9±24.8 vs. 11.7±24.5, respectively).

## Discussion

The World Health Organization declared SARS-CoV-2 a global pandemic on March 11, 2020; national lockdown in Spain was implemented three days later. At our center, ED consults and admissions for COVID19 infection rose progressively until they peaked the last week in March. Coincidentally, consults for other urgent pathology declined, and overall ED volume was reduced [5]. The peak was associated with >80% occupation of intensive care and 50% occupation of general floor beds with COVID19 patients. As of this writing, late March and early April continue to represent the maximum point of COVID19 hospitalizations in our region and center.

The objective of this study was not to evaluate costs specifically associated with COVID19 admissions but, rather, those associated with other urgent disease processes displaced or delayed for whatever reason by COVID19. Based on this study's findings, ED consults and admissions to surgery declined by up to nearly 50% in early 2020 relative to the same period in 2019, and presentations were delayed by an average of three days. Nonetheless, mean per-patient costs of hospitalization (including index admission and readmissions) were no different during the peak of the pandemic relative to the same period in 2019.

A logical assumption is that patients arriving later would have arrived sicker. The concept of fewer but sicker patients has not been borne out in this study in terms of the complications, mortality, and costs observed. Costs have been seen to be associated with the number and

**Table 2. Data related to hospitalization.**

|  | 2019 (N = 426) | 2020 (N = 346) | *P* |
|---|---|---|---|
| *Admission* | | | |
| Diagnostic category | | | 0.01 |
| Appendicitis | 106 (24.9) | 71 (20.6) | |
| Biliary tract pathology | 89 (20.9) | 103 (29.8) | |
| Gastrointestinal obstruction | 73 (17.1) | 49 (14.2) | |
| Non-obstructive gastrointestinal pathology | 85 (20) | 53 (15.4) | |
| Skin & soft tissue infection | 21 (4.9) | 25 (7.2) | |
| Abdominal trauma | 12 (2.8) | 3 (0.9) | |
| Readmission | 31 (7.3) | 28 (8.1) | |
| Other | 9 (2.1) | 13 (3.8) | |
| Onset of symptoms to diagnosis (days) | 2±3 | 5±22 | 0.01 |
| Non-operative management | 174 (41) | 150 (43) | 0.43 |
| Failure of non-operative management | 11 (7) | 18 (11) | 0.22 |
| *Surgical intervention* | | | |
| Intraoperative complications | 8 (2) | 5 (1) | 0.64 |
| Operative time (min) | 89±93 | 92±59 | 0.63 |
| *Hospitalization & outcomes* | | | |
| ICU stay (days) | 0.9±5 | 0.6±3 | 0.40 |
| Overall hospital stay (days) | 9±16 | 7±11 | 0.19 |
| Surgical reintervention | 12 (3) | 8 (2) | 0.66 |
| CCI | 10.3±23.7 | 13.9±25.5 | 0.10 |
| Mortality | 13 (3) | 13 (4) | 0.59 |

Variables related to admission and management and outcomes of hospitalization for patients with unplanned general surgery admissions during two periods: 02-05/2019 (pre-pandemic) and 02-05/2020 (COVID19 pandemic). Continuous variables are presented as mean ± standard deviation and categorical variables as "N" (%). CCI, Comprehensive Complication Index; ICU, intensive care unit.

severity of complications arising in the course of a disease process [15, 16]. Fewer general surgery patients were attended and admitted from the ED between February and May 2020 relative to the same period in 2019, yet those that were seen in 2020 did not present higher level of acuity than those seen in the period without pandemic.

We considered that while average costs might have been globally the same for the four-month periods under evaluation, a more detailed analysis comparing individual months might reveal differences. As reflected in Fig 2, March 2020 was the month with the lowest per-person healthcare costs. The decline in March, however, was not followed by any increase in costs in April or May 2020. There was no rebound effect in the opposite direction following the initial reduction in usage of emergency general surgery services.

While total per-person costs did not vary between periods under evaluation, some marginal but significant differences were observed in adjusted costs associated with consumable materials. While small, what these differences may reflect are changes in practice patterns in 2020 relative to 2019 (using, for example, more personal protective equipment in 2020) as opposed to higher degree of medical or surgical acuity, as clinical outcomes did not vary.

Considering diagnostic subpopulations, admissions for biliary tract pathology were similar if not slightly higher in 2020 relative to 2019. This observation appears consistent with what is known about dietary and other lifestyle changes that occur(ed) during the pandemic that are known to exacerbate biliary tract disease [17–19]. It may be the case that there was some decline in "appendicitis" cases based on fewer presentations of right lower quadrant pain that

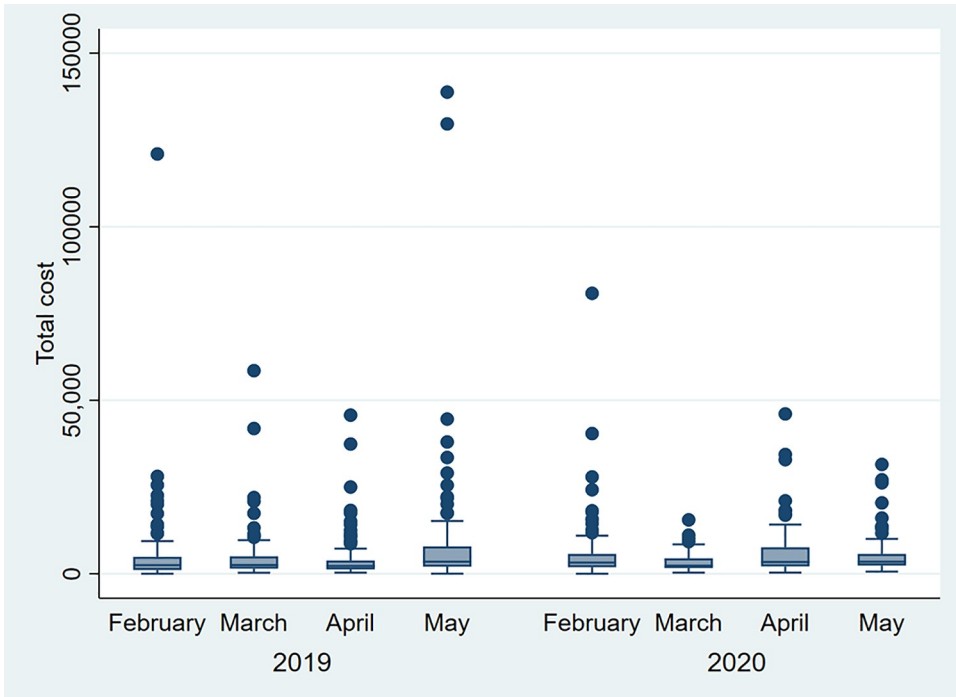

**Fig 2. Costs for emergency general surgery admissions.** Monthly evolution of total per-person healthcare costs for unplanned general surgery admissions during two periods: 02-05/2019 (pre-pandemic) and 02-05/2020 (COVID19 pandemic).

ultimately resolved with medical management. Given that this study did not impute outpatient costs, it is plausible that reduction in ED services for other diagnostic subsets was compensated to a degree by a shift toward greater utilization of outpatient resources. Outpatient care, however, appears to have experienced a similar decline as urgent/emergent consults. Healthcare in general was lost during the height of the pandemic in various parts of the world [1, 20, 21]. It Whether loss of healthcare for non-COVID19 patients accounts for some proportion of excess mortality observed in different countries, including Spain, where excess mortality between February and May was 38% [22], is conceivable but difficult to assess.

We attempted to evaluate cost differences in different diagnostic subpopulations. It is unclear whether higher costs for appendicitis in 2020 relative to 2019 were truly related to period or were a consequence of confounding variables, as cost adjustments could not be performed. The fact that complications for appendicitis admissions were no different in 2020 relative to 2019 is indicative of the latter.

Based on the reduction in unplanned general surgery admissions and lack of simultaneous increase in per-patient costs for hospitalization, we estimate that healthcare spending at our center was reduced by nearly 650,000€ for this particular population between February and May 2020 relative 2019. General surgery is responsible for 12–13% of our emergency department volume, not to mention elective activity. We can extrapolate further by considering the sum of similarly resource-intensive areas of medicine and reasonably assume that millions of Euros—potentially in excess of 5–6 million—in healthcare payments were lost at our institution during the height of the pandemic. Some if not much of this money may have been reallocated to the treatment of COVID19 patients, though expenditures for surgical and COVID19 care are not entirely interchangeable. The ultimate repercussions of these findings depend on the healthcare structure and reimbursement system and may be considered to include

**Table 3. Hospitalization and surgical costs.**

| | Unadjusted | | | | | | | Adjusted | | | | |
| | 2019 (N = 409) | | | 2020 (N = 334) | | | *P* | 2019 | | 2020 | | *P* |
| | Mean | SD | Median | Mean | SD | Median | | Mean | SE | Mean | SE | |
|---|---|---|---|---|---|---|---|---|---|---|---|---|
| *Total hospital-associated healthcare costs*[1,2] | 5,886.29 | 12,576.33 | 2,535.77 | 5,287.62 | 7,220.16 | 3,217.4 | 0.43 | 4,656.89 | 390.53 | 4,938.54 | 406.55 | 0.28 |
| *Hospitalization costs* | | | | | | | | | | | | |
| Pharmacy[1,3] | 401.16 | 2,034.31 | 22.89 | 241.76 | 877.82 | 37.73 | 0.17 | 177.98 | 79.06 | 184.77 | 88.10 | **0.04** |
| Transfusions[4] | 54.95 | 302.57 | 0 | 40.66 | 210.75 | 0 | 0.46 | 53.83 | 13.15 | 41.34 | 10.54 | 0.62 |
| Artificial implants[3] | 63.93 | 356.76 | 0 | 81.85 | 376.48 | 0 | 0.50 | 58.33 | 14.30 | 67.24 | 21.24 | 0.51 |
| Miscellaneous consumable materials[3] | 69.36 | 371.22 | 0 | 172.30 | 502.56 | 0 | **0.001** | 82.57 | 28.64 | 190.78 | 32.69 | **0.02** |
| Physician medical attention[3] | 380.27 | 1082.06 | 154.17 | 359.07 | 562.51 | 184.26 | 0.74 | 225.57 | 39.88 | 284.64 | 25.66 | 0.91 |
| General floor stay[3] | 2,319.95 | 5,236.91 | 866.01 | 2,021.07 | 3,326.40 | 992.45 | 0.36 | 1,444.76 | 238.67 | 1,485.71 | 148.55 | 0.34 |
| Studies and procedures[3] | 309.09 | 890.36 | 0 | 335.66 | 834.68 | 0 | 0.67 | 331.57 | 50.32 | 264.13 | 42.54 | 0.29 |
| Laboratory tests[2] | 85.78 | 277.83 | 13.17 | 79.16 | 150.31 | 25.76 | 0.69 | 56.12 | 9.44 | 70.21 | 10.60 | 0.48 |
| Specialist consultations[2] | 97.72 | 199.36 | 0 | 101.13 | 162.34 | 0 | 0.80 | 74.49 | 7.17 | 89.41 | 11.23 | 0.16 |
| Sanitation/janitorial services[3] | 150.51 | 267.76 | 64.50 | 127.99 | 169.80 | 73.72 | 0.17 | 100.54 | 12.67 | 99.90 | 8.17 | 0.09 |
| *Surgical costs* | | | | | | | | | | | | |
| OR staff[3] | 567.56 | 857.54 | 459.9 | 549.67 | 736.83 | 471.75 | 0.76 | 569.09 | 38.33 | 618.18 | 43.31 | 0.70 |
| OR usage[3] | 755.91 | 1,099.05 | 698.42 | 635.85 | 867.52 | 607.52 | 0.09 | 755.87 | 47.59 | 715.59 | 46.18 | 0.07 |
| OR consumables[3] | 200.44 | 409.52 | 198.11 | 153.56 | 180.11 | 198.11 | **0.04** | 199.02 | 17.34 | 169.73 | 9.47 | **0.01** |

Unadjusted and adjusted healthcare costs (2020 €) for patients with unplanned general surgery admissions during two periods: 02-05/2019 (pre-pandemic) and 02-05/2020 (COVID19 pandemic). ASA, American Society of Anesthesiologists; OR, operating room; SD, standard deviation; SE, standard error.

[1] Ajdusted costs calculated using a generalized linear model (GLM). All other costs were calculated using a two-part model, with GLM in the second part of the model.

[2] Adjusted costs adjusted by age, level of education, profession, ASA classification, and history of previous surgery.

[3] Adjusted costs adjusted by age, ASA classification, and history of previous surgery, based on collinearity.

[4] Adjusted costs adjusted by age and history of previous surgery.

everything from risk of healthcare-related job losses and loss of income for third parties providing healthcare-related goods and services to considerable financial profits for healthcare insurance providers due to unspent premiums [23].

This study has several limitations. Results we present are from the first wave of the pandemic in Spain and are influenced by the specific lockdown measures that were implemented at that time. As well, it considers resource utilization and costs in emergency general surgery only and does not evaluate other areas of healthcare. A study from Asia examined orthopedic surgery charges and found similar if not reduced in-hospital costs for orthopedic procedures (both urgent and elective) during the pandemic relative to pre-pandemic [24]. Another study from China regarding acute ischemic stroke, however, observed the opposite effect, with higher hospitalization costs during the height of the pandemic there [25]. Based on these findings, further cost studies in other areas of clinical medicine, in particular those requiring direct patient intervention and other activities that cannot be completed via telemedicine, are encouraged.

## Conclusions

During the height of the first wave of the COVID19 pandemic, in spite of close to 50% reduction in emergency general surgery consults and admissions, no differences in outcomes or per-patient healthcare costs were observed at our center. It is improbable that pathology truly disappeared, and this decline in activity undoubtedly represents part of the "untold burden" of patients that failed to seek care. As the pandemic continues, it is important to document not

only the health-related effects but also the economic consequences of loss of healthcare, which have important implications for patients, providers, payers, other healthcare stakeholders, and society.

## Acknowledgments

The authors would like to thank Ms. Adela Mas for assistance with database preparation, Ms. Elena Ramentol for assistance with data collection, and Dr. Luz María Peña-Longobardo and Dr. Ana Magdalena Vargas for their guidance with the statistical analysis.

## Author Contributions

**Conceptualization:** Amelia J. Hessheimer, Marta Trapero-Bertran, Antonio María de Lacy, Constantino Fondevila.

**Data curation:** Amelia J. Hessheimer, Alex Borin, Eugenia Butori, Anna Curell, Arlena Sofía Espinoza, Joaquín Jensen, Víctor Turrado, Xavier Morales.

**Formal analysis:** Marta Trapero-Bertran.

**Investigation:** Amelia J. Hessheimer, Alex Borin, Eugenia Butori, Anna Curell, Arlena Sofía Espinoza, Joaquín Jensen, Víctor Turrado, Xavier Morales.

**Methodology:** Amelia J. Hessheimer, Marta Trapero-Bertran, Antonio María de Lacy, Constantino Fondevila.

**Project administration:** Antonio María de Lacy, Constantino Fondevila.

**Supervision:** Antonio María de Lacy, Constantino Fondevila.

**Writing – original draft:** Amelia J. Hessheimer, Marta Trapero-Bertran.

**Writing – review & editing:** Alex Borin, Eugenia Butori, Anna Curell, Arlena Sofía Espinoza, Joaquín Jensen, Víctor Turrado, Xavier Morales, Antonio María de Lacy, Constantino Fondevila.

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
