## [Decision Letter · Decision Letter 0]

18 May 2021

PONE-D-21-12253

RESOURCE UTILIZATION AND OUTCOMES IN EMERGENCY GENERAL SURGERY DURING THE COVID19 PANDEMIC: AN OBSERVATIONAL COST ANALYSIS

PLOS ONE

Dear Dr. Hessheimer,

Thank you for submitting your manuscript to PLOS ONE. After careful consideration, we feel that it has merit but does not fully meet PLOS ONE’s publication criteria as it currently stands. Therefore, we invite you to submit a revised version of the manuscript that addresses the points raised during the review process.

Please revise accordingly.

We look forward to receiving your revised manuscript.

Kind regards,

Academic Editor

PLOS ONE

Journal Requirements:

3)  We note that you have indicated that data from this study are available upon request. PLOS only allows data to be available upon request if there are legal or ethical restrictions on sharing data publicly. For information on unacceptable data access restrictions, please see http://journals.plos.org/plosone/s/data-availability#loc-unacceptable-data-access-restrictions.

4)  Thank you for stating the following in the Competing Interests section:

[AJH and CF have received consultancy fees from Guanguong Shunde Innovative

Design Institute, Guangdong, China. The remainder of the authors have no

disclosures.].

5) Your ethics statement should only appear in the Methods section of your manuscript. If your ethics statement is written in any section besides the Methods, please delete it from any other section.

Reviewers' comments:

Reviewer's Responses to Questions

**Comments to the Author**

1. Is the manuscript technically sound, and do the data support the conclusions?

Reviewer #1: Yes

Reviewer #2: Yes

2. Has the statistical analysis been performed appropriately and rigorously? 

Reviewer #1: Yes

Reviewer #2: I Don't Know

3. Have the authors made all data underlying the findings in their manuscript fully available?

Reviewer #1: Yes

Reviewer #2: Yes

4. Is the manuscript presented in an intelligible fashion and written in standard English?

Reviewer #1: Yes

Reviewer #2: No

5. Review Comments to the Author

Reviewer #1: This is a nice manuscript about research and analysis about costs curtailed due to decreased number of surgical consults and procedures during the covid-19 pandemic. It is a plain expected fact due to lock downs affecting the elective as well as emergency surgical procedures. The amount saved in this segment of healthcare must have been spent and even much more further allocated to the care of covid-19 patients. I believe this comparison should also be part of the discussion.

Reviewer #2: 1) Useful study to evaluate the cost analysis and outcome of the surgical diseases during this pandemic.

2) This article needs english revision before publication. Length of the sentences should be minimised and easy to understand.

6. PLOS authors have the option to publish the peer review history of their article (what does this mean?). If published, this will include your full peer review and any attached files.

Reviewer #1: No

Reviewer #2: **Yes: **Dr. P. Jayakumar

---

## [Author Response · Author response to Decision Letter 0]

18 May 2021

May 18, 2021

Barcelona, Spain

Editorial Board, PLOS ONE

Dear Sir/Madam:

Thank you for reviewing our manuscript “RESOURCE UTILIZATION AND OUTCOMES IN EMERGENCY GENERAL SURGERY DURING THE COVID19 PANDEMIC: AN OBSERVATIONAL COST ANALYSIS” for publication in PLOS ONE. On behalf of our co-authors, we would like to thank you for the constructive comments that you have provided. Below, we provide a point-by-point response to the editorial team and each of the external reviewer’s specific comments:

Reviewer #1: This is a nice manuscript about research and analysis about costs curtailed due to decreased number of surgical consults and procedures during the covid-19 pandemic. It is a plain expected fact due to lock downs affecting the elective as well as emergency surgical procedures. The amount saved in this segment of healthcare must have been spent and even much [further] allocated to the care of covid-19 patients. I believe this comparison should also be part of the discussion.

We thank you for this astute comment and have provided commentary in this regard in the Discussion.

Reviewer #2: 1) Useful study to evaluate the cost analysis and outcome of the surgical diseases during this pandemic. 2) This article needs English revision before publication. Length of the sentences should be minimised and easy to understand.

Thank you for both of these comments; length of sentences has been reduced throughout the text.

As well, as per PLOS ONE specifications, we have reviewed the reference list and revised the text to meet PLOS ONE’s style requirements.

Regarding study data availability, we have now uploaded the minimal anonymized data set to the Harvard Dataverse:

https://dataverse.harvard.edu/dataset.xhtml?persistentId=doi:10.7910/DVN/UAAX8W

We have also modified the competing interests, as follows: “AJH and CF have received consultancy fees from Guanguong Shunde Innovative Design Institute, Guangdong, China. This does not alter our adherence to PLOS ONE policies on sharing data and materials. The remainder of authors has no disclosures.”

Finally, the ethics statement only appears in “Patients and methods” and has been deleted from all other sections.

We thank you again for your time and careful consideration of our manuscript. We hope you find the changes we have made suitable and may now consider it for publication in PLOS ONE.

Sincerely,

Amelia J. Hessheimer, MD, PhD

Marta Trapero-Bertrán, PhD

Antonio M. de Lacy, MD, PhD

Constantino Fondevila, MD, PhD

---

## [Decision Letter · Decision Letter 1]

26 May 2021

RESOURCE UTILIZATION AND OUTCOMES IN EMERGENCY GENERAL SURGERY DURING THE COVID19 PANDEMIC: AN OBSERVATIONAL COST ANALYSIS

PONE-D-21-12253R1

Dear Dr. Hessheimer,

We’re pleased to inform you that your manuscript has been judged scientifically suitable for publication and will be formally accepted for publication once it meets all outstanding technical requirements.

Kind regards,

Academic Editor

PLOS ONE

Additional Editor Comments (optional):

Reviewers' comments:

Reviewer's Responses to Questions

**Comments to the Author**

1. If the authors have adequately addressed your comments raised in a previous round of review and you feel that this manuscript is now acceptable for publication, you may indicate that here to bypass the “Comments to the Author” section, enter your conflict of interest statement in the “Confidential to Editor” section, and submit your "Accept" recommendation.

Reviewer #1: All comments have been addressed

Reviewer #2: All comments have been addressed

2. Is the manuscript technically sound, and do the data support the conclusions?

Reviewer #1: Yes

Reviewer #2: Yes

3. Has the statistical analysis been performed appropriately and rigorously? 

Reviewer #1: Yes

Reviewer #2: I Don't Know

4. Have the authors made all data underlying the findings in their manuscript fully available?

Reviewer #1: Yes

Reviewer #2: Yes

5. Is the manuscript presented in an intelligible fashion and written in standard English?

Reviewer #1: Yes

Reviewer #2: Yes

6. Review Comments to the Author

Reviewer #1: Thank you for addressing the minor points raised in the original submitted manuscript. It is now appropriate.

Reviewer #2: Useful study to evaluate the cost analysis and outcome of the surgical diseases during this pandemic.

7. PLOS authors have the option to publish the peer review history of their article (what does this mean?). If published, this will include your full peer review and any attached files.

Reviewer #1: No

Reviewer #2: **Yes: **Dr. P. Jayakumar

---

## [Editor Report · Acceptance letter]

10 Jun 2021

PONE-D-21-12253R1 

Resource utilization and outcomes in emergency general surgery during the COVID19 pandemic: An observational cost analysis 

Dear Dr. Hessheimer:

I'm pleased to inform you that your manuscript has been deemed suitable for publication in PLOS ONE. Congratulations! Your manuscript is now with our production department. 

Kind regards, 

on behalf of

Dr. Robert Jeenchen Chen 

Academic Editor

PLOS ONE